# Activation Control for Efficiently Eliciting Long Chain-of-thought Ability of Language Models

**Zekai Zhao**[∗]**, Qi Liu**[∗]**, Kun Zhou**[†]**, Zihan Liu, Yifei Shao, Zhiting Hu, Biwei Huang**
University of California, San Diego.
`kuzhou@ucsd.edu`

## Abstract

Despite the remarkable reasoning performance, eliciting the long chain-of-thought (CoT) ability in large language models (LLMs) typically requires costly reinforcement learning or supervised fine-tuning on high-quality distilled data. We investigate the internal mechanisms behind this capability and show that a small set of high-impact activations in the last few layers, greatly govern the long-form reasoning attributes, *e.g.,* output length and self-reflection. Through simply amplifying these activations and adding "wait" tokens, the long CoT ability can be invoked without training, leading to significantly increased self-reflection rate and accuracy. In addition, we also find that the activation changes follow predictable trajectories, *i.e.,* a sharp rise after special tokens and a subsequent exponential decay. Based on these insights, we introduce a general training-free activation control technique. It utilizes a few contrastive examples to identify the relevant activations, and then incorporates simple analytic functions to adjust their values at inference time to elicit long CoTs. Extensive experiments have verified the effectiveness of our methods in efficiently eliciting the long CoT ability of LLMs and improving the performance. Besides, we further propose a parameter-efficient fine-tuning method that trains only the last-layer activation amplification module and a few LoRA layers, outperforming LoRA on reasoning benchmarks with much fewer parameters. Our code and data are fully public released at `https://github.com/ZekaiZ123/EELo-CoT`.

## 1 Introduction

On the path to artificial general intelligence (AGI), enhancing the reasoning ability of large language models (LLMs) [1, 2, 3, 4] remains to be one of the most important challenges. Techniques like chain-of-thought (CoT) prompting [5] can elicit good performance on reasoning tasks by prompting the LLM to generate intermediate reasoning steps. However, in complex reasoning tasks (*e.g.,* math competition problems), such a way is prone to make mistakes in intermediate steps, finally failing to reach the accurate answer. Recently, long-CoT models and systems such as OpenAI-o1 [3] and DeepSeek-R1 [6] have exhibited remarkable performance in solving complex tasks. These methods can perform human-like slow-thinking behaviors, with quite a few careful deliberation and self-reflection steps before generating the final answer [6].

To elicit the long CoT ability, existing work relies on either reinforcement learning (RL) on high-quality instances with proper annotations, or supervised fine-tuning (SFT) on distilled data [7, 8, 9]. However, in practice, it is costly to collect enough well-annotated high-quality instances or distilled data [8], especially for complex reasoning tasks that humans may also not perform well. Besides, the RL process is also complicated, and requires expert knowledge to control the training stability and

---

[∗]Equal Contribution
[†]Corresponding Author

39th Conference on Neural Information Processing Systems (NeurIPS 2025).

effectiveness [10, 11]. In fact, the key bottleneck comes from the lack of understanding of *how the long CoT ability is elicited in LLMs*. Since LLMs have undergone a very long pre-training stage and then a relatively short post-training stage using RL or SFT, it is possible that the long-CoT ability already exists in base models [6, 12]. If so, we can efficiently wake up the long-CoT ability when necessary, and exert influence to achieve more fine-grained controls in base models.

In LLM, activation values of MLP layers play a key role in preserving key information or styles during inference [13, 14]. Inspired by it, we analyze and compare the activation dynamics in state-of-the-art LLMs with and without the long-CoT ability during inference on complex reasoning tasks. By tracing the layer-wise activations, we identify specific positions and patterns that correlate with desirable CoT traits (*e.g.,* longer and with self-reflection). These activations generally locate in the last few layers (see Fig. 1). By simply amplifying them and adding the "wait" token, the long CoT ability can be invoked without training (see Fig. 3). In addition, we also find that the activation changes in these positions follow predictable patterns, *i.e.,* a sharp rise after the trigger tokens (*e.g.,* wait) and a subsequent exponential decay (see Fig. 4). Therefore, it is promising to utilize an analytic function to fit the pattern and mimic the activation to efficiently elicit the long-CoT ability. Based on the above findings, we propose a training-free activation control approach, to elicit the long-CoT ability in the inference process. Concretely, we first collect a few pairs of contrast examples about the long-CoT ability. Then, we use the contrast examples to identify the related activation positions and collect patterns in the base model. Next, we formulate the pattern of each activation value into a unified function with few coefficients to control the change tendency and intervention intensity. Based on the collected patterns, we fit these coefficients, and devise a simple rule that triggers intervention after encountering special tokens during inference.

Since we do not need to train the LLM and the activation control method only requires using base models, our approach is general to any LLMs and any datasets for efficiently elicit the long-CoT ability when necessary. Extensive experiments conducted on complex math and science reasoning benchmarks have demonstrated the effectiveness of our approach in improving the LLM performance (see Table 3) with special long-CoT style. We also reveal that the long-CoT related activation positions become more inactive or even dead in over-posttrained LLMs.

Furthermore, as we can identify the key activations within the LLM, we also propose a parameter-efficient fine-tuning method that focuses on automatically learning the activation pattern from long-CoT related training data. Concretely, we add a rather lightweight learnable amplification module for the identified activations in the last layer, and add LoRA layers with a low rank in the former layers, while fixing the parameters of other parameters. In this way, our approach only requires training much fewer parameters than the existing LoRA method. Experimental results have shown that our approach can achieve better performance than LoRA. It achieves comparable performance of full-parameter fine-tuning method, but only trains 1.51% of total parameters.

## 2    Empirical Analysis on the Activation Patterns of Long CoT Ability

In this section, we empirically analyze the activation patterns underlying the long CoT reasoning ability, since existing work has shown that activation values can preserve key information of styles or concepts [13]. We conduct a series of experiments on state-of-the-art Qwen2.5-7B series LLMs, and first study (1) the distribution of long-CoT related activations, then test (2) whether activation intervention can elicit long-CoT ability, finally analyze (3) whether there are predictable dynamic patterns in activation values during inference.

### 2.1    Long-CoT Related Activations Distribution Study

We aim to reveal how the key activation values are distributed within an LLM and the difference between LLMs with and without the long-CoT ability. We first collect a set of contrastive example pairs about the long-CoT characteristics, which are fed to the LLM to collect the long-CoT related activations. Then, we analyze the distribution of the top ones across layers and different LLMs.

**Analysis Setup.**    We conduct the experiments on MATH [15], a widely used mathematical reasoning dataset. For the contrastive example pairs, we consider the following three key characteristics of long-CoT ability, *i.e.,* longer, with self-reflection, and more accurate. Thus, we build two datasets that contain positive and negative samples that satisfy and unsatisfy the above features, respectively.

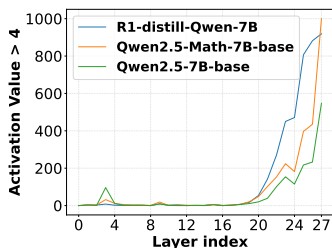
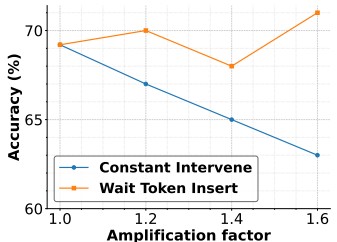
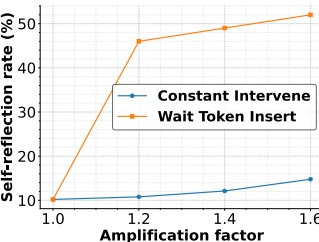

Figure 1: (a) Sparse Activations when processing Long CoT

Figure 2: (b) Model Accuracy and Amplification Scale

Figure 3: (c) Self-Reflection Ratio and Amplification Scale

Concretely, we randomly sample 160 questions from the training set of MATH dataset and generate CoT responses using two models, *i.e.,* R1-distilled-Qwen-7B [6] and Qwen-2.5-7B-Instruct [16]. As the former one has acquired the long-CoT ability distilled from DeepSeek-R1 while the last one not, we sample the positive and negative ones from their responses, respectively. Then, we feed each question with corresponding positive and negative responses into three LLMs (*i.e.,* R1-distilled-Qwen-7B, Qwen2.5-7B-Math-base, and Qwen2.5-7B-base), and compute the activation value difference in all the MLP layers. Finally, we average all the activation value difference, and select the ones with higher difference (*i.e., $> 4$*) as the long-CoT ability related activations.

**Finding-1: Long-CoT Activations Mainly Exist in Last Few Layers.** As shown in Fig. 1, the vertical axis indicates the number of activations whose activation values greater than 4. Only very few long-CoT related activations exist in the former layers, *i.e.,* near zero before the 18-th layer. After that, the number of long-CoT activations increases in a near-linear tendency, and the last layer even contains more than 50% long-CoT activations of the LLM. It indicates that the last few layers contribute more on the long-CoT ability.

**Finding-2: Long-CoT LLM Contains More Long-CoT Related Activations.** By comparing the curves of R1-distilled-Qwen-7B with other two LLMs, we can see its long-CoT related activations are consistently more than the two models. Thus, we have the hypothesis that activation matters in eliciting the long-CoT ability, and will verify it then.

## 2.2 Ability Control through Activation Intervention Study

We study whether the related activations found can be used to elicit the long-CoT ability. We adopt a rather simple way that consistently amplifies the values of these activations and adds "wait" token during inference, and observe whether the accuracy and self-reflection rate can also increase.

**Analysis Setup.** We first use the above contrastive example pairs to measure the long-CoT ability correlation of each activation. Then, we rank all the activations, and select the top-200 ones as the key activations. For intervention, we use the following amplification factors, *i.e.,* 1.2, 1.4, and 1.6, and larger factors would cause the generation unstable. In our experiments, we see that only amplifying the activations can not effectively lead to stable self-reflection action. Thus, we use a simple way that inserts the "wait" token at the start of the sentence once the last generated sentence contains the math equation. It serves as a trigger token to force LLMs to perform self-reflection. We visualize the accuracy and self-reflection rate in the test set of MATH dataset. The self-reflection rate is computed by computing the percentage of responses that contain special reflection tokens and phrases, *e.g.,* "wait" and "let me double check".

**Finding-3: Simple Activation Amplification with Wait Token Insert Induces Long-CoT Reasoning.** As shown in Fig. 2 and Fig. 3, amplifying the activation value with the wait token insert strategy can significantly improve both the self-reflection rate and accuracy. Actually, combining activation amplification with the insertion of the "wait" token yields greater improvements than using activation amplification alone. As the Table 1 shown below, while inserting the wait token alone does increase the self-reflection rate, its effect on accuracy is limited. In contrast, the combination of activation amplification and wait token insertion leads to a more substantial and consistent improvement in both accuracy and self-reflection. This supports our claim that the two components are complementary.

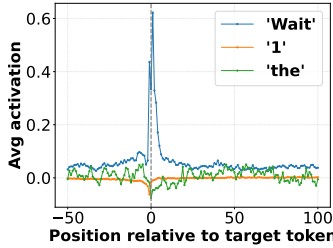 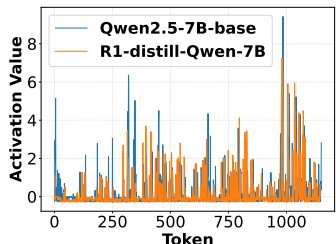 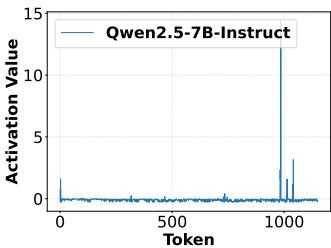

Figure 4: (a) Wait Token Insert Induces Long-CoT Reasoning

Figure 5: (b) Activation Patterns of base and long CoT LLMs

Figure 6: (c) Activation Pattern of Qwen2.5-7B-Instruct

To better show the comparison of the two methods, we randomly select one example in GPQA shown in Table 9. We can see that although wait-token-only enforces the LLM to perform "self-reflection", but the generated contents are just repeating the question. It is just like an imitation of reflection without thinking. In comparison, with the activation control strategy, the LLM starts to think deeply and check the correction of the answer in the self-reflection parts.

Table 1: Inference on GPQA benchmark

| Model Name | Add Wait Token Only | Add Token and Activation Values Applied |
|---|---|---|
| Qwen 2.5 7B | 33.33% Acc | 35.86% Acc |
| Qwen 2.5 Math 7B | 32.83% Acc | 37.88% Acc |

Therefore, it is promising to design a more systematic approach by utilizing the coordination between wait-token-insertion and activation intervention, to efficiently elicit long-CoT ability.

## 2.3 Activation Dynamics Analysis

To help design appropriate trigger token and activation intervention strategies, we further conduct the qualitative study to analyze the activation dynamics of different LLMs. We aim to find out whether each activation has a special pattern that can be predictable during inference.

**Analysis Setup.** For analyzing the activation dynamics, we track the inference-time value changes of the top-1 activation (found using contrastive example pairs) from Qwen2.5-7B-base, Qwen2.5-7B-Instruct, and R1-distill-Qwen-7B. We randomly select few questions from MATH test set, and have seen very similar tendency across them. Thus, we visualize a random one in Fig. 5. Besides, we also see that the high activation values often appear after the "wait" token during inference. Thus, we collect the activation values and their relative positions to few special tokens, and then draw the figure to visualize the different activation change tendencies around different special tokens.

**Finding-4: Base and Long-CoT Models Exhibit Similar Sparse Activation Dynamics.** As shown in Fig. 4, the top-1 activation of the two LLMs is activated with a very sparse pattern during inference. It is mostly near zero value, but activated into a relatively high value in special positions (*e.g.,* "wait" token). We see that the high-value positions of the two LLMs are mostly the same ones, and the corresponding values are also very similar. It indicates that the base and long-CoT models have similar sparse activation dynamics.

Based on the above findings, it is promising to use the base LLM itself to predict the long-CoT related activations.

**Finding-5: Instruct Model Activations are Very Inactive.** As shown in Fig. 6, the activations of the instruct model are mostly near zero. In contrast to the base model, it is very inactive and even like "dead" activations. A possible reason is that these activations have been biased after learning large-scale short instructions in the post-training stage. This may make it hard for the LLM to adapt to this new reasoning pattern, as shown by other attempts to elicit the long-CoT ability of LLMs [17, 6].

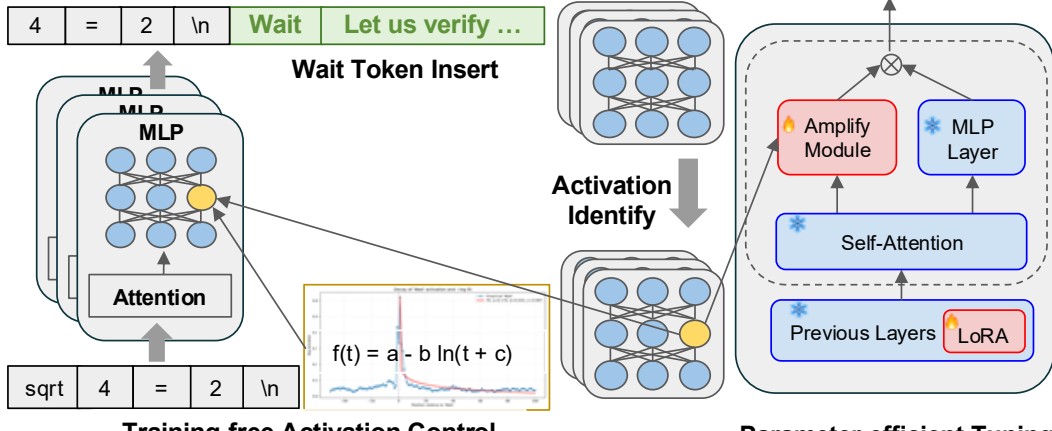

Figure 7: The overall framework of the proposed Long-CoT elicitation method. Based on the identified activations, the left part is the proposed training-free activation control method, and the right part is the parameter-efficient training method.

**Finding-6: Activations around Wait Token Have Predictable Pattern.** As shown in Fig. 4, the activation value curves around the common token "the" and digital token "1" both contain very few changes, while the one of the wait tokens has a rather outstanding sharp rising then falling tendency. It indicates that the wait token is a special trigger token that the long-CoT LLM has learned for waking up the activation action. Besides, the falling part of the wait token curve is likely to follow a logarithmic decay function.

Figure 4 is indeed designed to qualitatively illustrate the contrast in activation values between reflective triggers (e.g., "wait") and neutral tokens (e.g., "the", digits). While the qualitative pattern is compelling, adding quantitative metrics would strengthen the evidence. Table 2 shows that 1.The average activation magnitude of long-CoT-related neurons when each token appears. 2.The proportion of selected neurons that exceed a threshold (e.g., activation > 4) for each token type. As shown, the "wait" token reliably elicits higher average activation and activates a greater fraction of long-CoT neurons compared to neutral tokens like "the" or mathematical operators like "equals". This provides quantitative confirmation that certain tokens (e.g., "wait") serve as interpretable triggers for switching the model into a reflective reasoning mode.

| Token | Avg Activation | %>0.4 |
|--------|----------------|-------|
| Wait | 0.816 | 5.3% |
| The | -0.1322 | 0.0% |
| Equals | 0.116 | 2.0% |

Table 2: Token Activation Statistics

As the above trigger word and activation patterns are very significant, it is promising to design a function to mimic them for eliciting the long-CoT ability of LLMs.

## 3    Training-free Activation Control Method

According to our findings in Section 2.1, the activation patterns of long-CoT ability are predictable. Therefore, it is feasible to efficiently elicit long-CoT reasoning through activation control. In this section, we develop a training-free activation control method, namely EELo-CoT, and evaluate its effectiveness on three complex reasoning tasks.

## 3.1 Methodology

Our method is composed of the analytic function based activation intervention and the forcing reflection strategies. Given the LLM, our method only requires few contrastive example pairs to help fit a function, and then can elicit it to perform long-CoT reasoning when necessary.

**Activation Amplification with Analytic Function.** According to Finding-6 in Section 2.3, the activation values associated with the long-CoT ability follow a distinct logarithmic decay function (*e.g.*, $f(t) = -\log(t)$). To capture this, we utilize the contrastive example pairs to identify key activations (following Section 2), and then collect the value trajectories of all activations across a fixed token window following the token "Wait". Next, we use the above data to fit the following function. Here, we show the computed coefficients of Qwen2.5-7B-base as an example:

$$f(t) = a - b \cdot \log(t + c) \quad \text{where} \quad a = 0.17, \ b = 0.033, \ c = -0.997 \tag{1}$$

As shown in Fig. 4, this curve can well capture the activation changes of Qwen2.5-7B-base. Based on it, we further design an activation amplification rule. Let $A$ denote the original activation value of the LLM at the current token, where $t$ is its relative distance with the trigger token. The new activation $A'$ is computed as:

$$A' = A \cdot (1 + \alpha f(t)), \tag{2}$$

where $f(t)$ produces the reference value of the amplification, and $\alpha$ is the tunable scaling factor.

**Forcing Reflection after Reasoning.** Based on the above activation intervention strategy, we devise the forcing reflection strategy to support it, for deliberating the last reasoning step if necessary. Concretely, we leverage the number of digits in the last sentence as the metric to determine if perform forcing reflection. Once we detect $k$ or more digits, we insert a "wait" token in the starting position of the next sentence. Then, the next sentence will continue to perform self-reflection after the "wait" token, and meanwhile the analytic function based intervention strategy will also be activated to guide the LLM.

In our observations, errors frequently arise when the model performs arithmetic near the end of reasoning chains—precisely where a self-reflection trigger like "wait" can help.

Thus, we adopt a simple trigger strategy: if the last sentence contains more than a threshold number of digits (e.g. 5), we inject a "wait" token before the next sentence, prompting the model to reconsider its previous steps.

In addition, we also add a cool-down window setting that temporarily locks down the forcing reflection strategy in the next four sentences after its execution. Such a way prevents the LLM from repeating meaningless self-reflection during inference.

## 3.2 Experimental Settings

We introduce the details of our experimental setting to evaluate the training-free method.

**Datasets.** We select the following three benchmarks for evaluation:

• *MATH* [15]: it consists of 500 high-school level math problems across algebra, geometry, calculus, and number theory.

• *AMC23*: it consists of problems from the 2023 American Mathematics Competitions (AMC 10 and AMC 12), covering challenging multi-choice problems designed for high school students.

• *GPQA-Diamond* [18]: GPQA benchmark focuses on high-complexity questions. We select the Diamond split that includes only the most difficult examples.

**Baselines.** In our results table, the Forcing Reflection row corresponds to the intervention where only the token "wait" is inserted to encourage reflective reasoning. Constant Intervention represents a baseline where only a fixed amplification factor is applied to selected neuron activations throughout the entire reasoning process. Forcing & Constant combines both the reflection-triggering token and the constant amplification strategy. Finally, Our Rule refers to our proposed intervention method, which applies a dynamic, activation-based scheduling function along with the token insertion rule to modulate neuron behavior.

Table 3: The evaluation results of our method using Qwen2-7B-base, Qwen2.5-7B-base, Qwen2.5-Math-7B-base on Math500, AMC23 and GPQA

| Scenarios | Math500 | | | AMC23 | | | GPQA | | |
|---|---|---|---|---|---|---|---|---|---|
| | Acc. | Length | Reflect | Acc. | Length | Reflect | Acc. | Length | Reflect |
| **Qwen2-7B-base** | 30.80 | 685.52 | 3.20 | 12.50 | 795.75 | 2.50 | 26.77 | 494.35 | 6.06 |
| + Forcing Reflection | 30.00 | 1019.13 | 65.20 | 10.00 | 1029.2 | 70.00 | 26.77 | 781.29 | 66.67 |
| + Constant Intervention | 28.60 | 761.64 | 3.40 | 7.50 | 729.83 | 7.50 | 28.28 | 484.92 | 5.56 |
| + Forcing & Constant | 29.20 | 990.91 | 65.00 | 20.00 | 1096.88 | **80.00** | 26.77 | 856.33 | 64.65 |
| **+ EELo-CoT (Ours)** | **35.60** | 938.74 | **66.20** | **20.00** | 1146.2 | 77.50 | **30.30** | 774.31 | **65.15** |
| **Qwen2.5-7B-base** | 69.20 | 328.20 | 10.20 | 45.00 | 436.15 | 7.50 | 30.30 | 457.34 | 4.04 |
| + Forcing Reflection | 66.00 | 376.75 | 47.80 | 40.00 | 613.33 | 62.50 | 33.33 | 598.06 | 68.69 |
| + Constant Intervention | 69.20 | 329.32 | 11.40 | 45.00 | 488.23 | 17.50 | 33.33 | 466.95 | 5.56 |
| + Forcing & Constant | 66.40 | 384.76 | 45.40 | 47.50 | 583.62 | **80.00** | 31.31 | 598.22 | **71.72** |
| **+ EELo-CoT (Ours)** | **73.00** | 369.25 | **49.60** | **57.50** | 443.52 | 70.00 | **35.86** | 585.78 | 68.18 |
| **Qwen2.5-Math-7B-base** | 68.00 | 381.67 | 73.80 | 65.00 | 547.70 | 60.00 | 33.84 | 476.88 | 28.28 |
| + Forcing Reflection | 64.00 | 424.32 | 88.40 | 57.50 | 549.50 | 90.00 | 32.83 | 650.62 | 78.79 |
| + Constant Intervention | 64.20 | 381.28 | 71.40 | 45.00 | 583.02 | 65.00 | 33.33 | 461.90 | 29.29 |
| + Forcing & Constant | 62.60 | 416.90 | 90.20 | 47.50 | 639.83 | **97.50** | 34.34 | 515.15 | 78.79 |
| **+ EELo-CoT (Ours)** | **76.00** | 441.00 | **90.60** | 65.00 | 625.50 | 95.00 | **37.88** | 552.49 | **78.79** |

**Implementation Details.** We randomly sample problems from the MATH training dataset and generate CoT responses using two models: the R1-distilled Qwen 7B and the Qwen 2.5 7B Instruct. We filter these into two groups: Group 1: Contains responses that show self-reflection (e.g., "wait", "let me double check"), exceed 1000 tokens, and are correct. Group 2: Contains responses with no self-reflection, shorter than 1000 tokens, and incorrect. After filtering, we obtain 160 matched question pairs. Each CoT is passed through a model while recording the MLP neuron activations. We compute the per-neuron activation differences between the two groups. The self-reflection ratio reflects the presence of self-reflective behaviors in the generated CoTs, which is computed by matching a curated list of reflective phrases (*i.e.,* "let me double check", "wait", "verify"). We set the minimum number of digits $k$ in the last sentence to insert "wait" token in the starting position of the next sentence as 5. The cool-down window setting that temporarily locks down the forcing reflection is set as 4. The number of activations we used are 150 with amplification factor set as 4.

## 3.3 Results Analysis

**Main Results.** The results of our method and baseline in our evaluation are presented in Table 3. We can observe that our method achieves the highest accuracy on both base models on Math500: 72.00% for Qwen2.5-7B-base and 76.00% for Qwen2.5-7B-Math-base, surpassing constant and trigger-based interventions by up to 6 percentage points. The self-reflection rate also increases substantially, reaching 90% with our rule compared to only 10.2% in the original base and 73.8% in the math-tuned model. Our method balances this with moderately increased output length. On the AMC23 benchmark, our rule also shows strong gains in reasoning quality. For Qwen2.5-7B-base, it improves accuracy from 45.00% to 57.50%, and self-reflection from 7.5% to 70.00%. On the math-tuned variant, our method achieves 65.00% accuracy and the highest reflection rate of 95.00%. GPQA presents a knowledge-intensive challenge. Our rule yields notable improvements, boosting accuracy from 30.30% to 35.86% on Qwen2.5-7B-base, and from 33.84% to 37.88% on Qwen2.5-7B-Math-base. In both cases, self-reflection rates increase to nearly 79%, confirming the rule's generalizability across reasoning domains.

**Applying to Other LLMs.** The results of our method and baseline tested on other LLMs are presented in Table 4. We can see that Qwen2.5-1.5B-base model achieved an Accuracy of 18.60 % compared to the baseline 15.60 % and at the same time reflection rate increases dramatically from 0.20% to 23.40%, suggesting that the intervention effectively triggers self-monitoring behavior. Measured on the GPQA, Accuracy improves from 22.73 % to 28.79 %, showing a consistent positive effect on complex, knowledge-intensive question answering. Reflection rate increases from 2.53 % to 34.34 %, again indicating a strong promotion of reflective behaviors, which may correlate with

Table 4: The evaluation results of our method using Qwen2.5-1.5B-base, Qwen2.5-3B-base

| Scenarios | Math500 | | GPQA | |
|---|---|---|---|---|
| | Accuracy | Reflection | Accuracy | Reflection |
| **Qwen2.5-1.5B-base** | 15.60 | 0.20 | 22.73 | 2.53 |
| **+EELo-CoT (Ours)** | **18.60** | **23.40** | **28.79** | **34.34** |
| **Qwen2.5-3B-base** | 27.80 | 0.80 | 27.27 | 5.05 |
| **+EELo-CoT (Ours)** | **31.20** | **33.40** | **28.28** | **27.27** |

improved reasoning depth. Similarly, Qwen2.5-3B-base model achieved higher accuracy than its baseline 27% and with a much higher reflection rate on Math500 and GPQA respectively.

# 4 Parameter-efficient Training Method

Since the long-CoT related activations are sparse and predictable, it is feasible to train a lightweight activation adaptation module specially for controlling these activations. As such a way is straightforward to learn the key of long-CoT ability, it can be more efficient and effective. In this section, we develop a parameter-efficient training method, which updates 1.51 % of the total parameters and can perform better than LoRA (update 6.15 % parameters) and even full-parameter trained models.

## 4.1 Methodology

In our method, we utilize few contrastive example pairs to help identify the key activations, and then train the activation amplification module with LoRA layers in the LLM.

**Activation Amplification Module.** Our activation amplification module is actually a special gate layer that can amplify the activation value within a certain range. In the typical MLP layer of LLMs, the activation value is computed by first multiplying the input vector $\mathbf{x} \in \mathbb{R}^h$ with a projection matrix $\mathbf{W}_g \in \mathbb{R}^{h \times d}$ and then feeding it into the activation function $e.g.,$ ReLU and SiLU.

$$A(\mathbf{x}) = \text{Act}(\mathbf{x}\mathbf{W}_g) \tag{3}$$

On top of this, we incorporate an activation amplifier module that consists of a linear projection matrix $\mathbf{W}_a \in \mathbb{R}^{h \times n}$ and a scaled sigmoid function, where $n$ denotes the number of identified key activations, $i.e.,$ 100. Given the input $\mathbf{x}$, we multiply it with the projection matrix $\mathbf{W}_a$, and feed it to the sigmoid function to normalize the value range to $[0, 1]$. Then, we multiply it with a scale coefficient $\beta$ to obtain the amplification scale vector for the $n$ activations. Next, we multiply the original activation values with the corresponding amplification scale in the vector. All the above modifications can be formulated as the following equation:

$$A(\mathbf{x}) = \text{Act}(\mathbf{x}\mathbf{W}_g) \odot \sigma(\mathbf{x}\mathbf{W}_a) \cdot \beta \tag{4}$$

where $\text{Act}$ is the activation function and $\sigma$ is the sigmoid function. In this way, this module enables adaptively control the amplification with the consideration of the input context.

**Parameter-efficient Training.** During training, we add the activation amplification module in the last layer, and also add LoRA in other layers. Since our empirical findings have shown that most long-CoT related activations are in the last few layers, we reduce the rank of the former layer LoRA to 64, while existing work sets the rank to 256 for ensuring the effectiveness [19]. Besides, since only $\mathbf{W}_a \in \mathbb{R}^{h \times n}$ and the scalar $\beta$ in the amplification module are trainable, this module is also more parameter-efficient than LoRA, $i.e.,$ $(h \times 100 + 1)$ vs. $(h + d) \times 256$.

## 4.2 Validation Experiments

**Experimental Settings.** To verify the effectiveness of our parameter-efficient training method, we fine-tune Qwen2.5-32B-Instruct on LIMO dataset [20]. LIMO contains 817 carefully curated training samples and is specifically designed to improve the model's mathematical and logical reasoning

Table 5: Performance comparison on Math500, AMC23, and GPQA benchmarks. Each benchmark includes both accuracy and average length.

| Method | %$_{\text{param.}}$ | Math500 | | AMC23 | | GPQA | |
|---|---|---|---|---|---|---|---|
| | | Accuracy | Length | Accuracy | Length | Accuracy | Length |
| Full Finetuning | 100 | **91.60** | 3642.71 | **92.50** | 14170.80 | 69.19 | 7770.40 |
| LoRA | 6.15 | **91.60** | 3952.61 | 85.00 | 14827.93 | 66.17 | 8508.25 |
| **Efficient Fine-tuning (Ours)** | **1.51** | 90.20 | 3754.20 | 88.75 | 7077.48 | **70.02** | 8593.46 |

abilities. For implementation, our method was evaluated against both LoRA and full-parameter fine-tuning baselines. For the LoRA baseline, we set the rank to 256 and the scaling factor $\alpha$ to 512, applying LoRA to all eligible layers in the model. In contrast, our method adopts a more parameter-efficient design by using a lower rank of 64 on the first 63 decoder layers. Additionally, we inject an Activation Amplification Module into the final MLP layer. All original model parameters, except parameters in LoRA and Amplification Module are frozen. The number of amplified key activations, $n$, is set to 100. Our fine-tuning process completed within 8 hours on 8 NVIDIA A100 GPUs. To ensure a fair comparison, we adhere to the evaluation protocol outlined in previous work [20], assessing all methods on three benchmarks: AMC23, GPQA, and Math500. Specifically, we use the pass@1 metric across all benchmarks under a zero-shot chain-of-thought (CoT) setting. For Math500 and GPQA, we evaluate correctness via greedy decoding with a single sample per question. For AMC23, we generate 16 samples per question with a temperature of 0.7 and compute the unbiased pass@1 metric as proposed by [21]. In addition to evaluating model accuracy on the benchmarks, we computed the average response tokens generated by models for each problem, including both the CoT reasoning process and the final answer. This metric is crucial for assessing inference efficiency. All evaluations are accelerated using vLLM [22] for efficient inference.

**Result Analysis.** Our method demonstrates that training only 1.51% of model parameters can achieve equivalent performance to that of full-parameter fine-tuning across all three benchmarks. This highlights the sparsity and localization of parameters responsible for complex reasoning behaviors such as long chain-of-thought (CoT) and self-reflection. This observation challenges the conventional assumption that large-scale fine-tuning is always necessary to acquire complex reasoning abilities. Instead, our approach shows that only a small, identifiable set of parameters can elicit long CoT ability. Although the model is trained solely on a small math-focused dataset, it attains 70.02% accuracy on the science-oriented GPQA benchmark, outperforming both the LoRA baseline (66.17%) and full fine-tuning (69.19%). This indicates that the selected activation patterns support reasoning strategies that generalize beyond the training domain. Moreover, on the AMC23 benchmark, our method reduces the average number of tokens used during inference by approximately 50%.

## 5 Related Work

**Large Language Models.** LLMs have demonstrated remarkable capabilities in a variety of NLP tasks [23, 24, 25], and the performance of cutting-edge models (*e.g.,* Gemini, Grok, and Qwen) has also developed rapidly [26, 27, 28]. Generally, LLMs are first pre-trained on large-scale unsupervised corpus, and then fine-tuned on instructions and human alignment data, to better adapt to various tasks and applications [29, 30]. To support more complex scenarios in real world, enhancing the reasoning ability is the key challenge for LLMs [3]. Early work mainly focuses on optimizing the prompt engineerings such as chain-of-thought and tree-of-thought [5, 31], or collect more reasoning-related data to fine-tune or continual pre-train the model [32, 33, 34, 35, 36]. However, collecting high-quality reasoning-related data is challenging, due to the difficult nature of reasoning problems. Other strategies are proposed to utilize tool augmentation [37], and non-training decoding optimization [38], to efficiently improve the reasoning performance. Recent directions from test-time search strategies [39, 31, 40] to RL-based training [41, 42] have shown that LLMs can be pushed toward more deliberate reasoning. However, the internal processes that give rise to reasoning ability are still poorly understood. Several interpretability studies attempt to pinpoint few reasoning-related neurons [43], or turns to representation-space manipulations to steer coordinated neuron activity [44].

**Long Chain-of-thought Ability.** Recent "slow-thinking" models, *e.g.,* OpenAI o1 [3], DeepSeek-R1 [6], and QwQ [2] have pushed the frontiers of LLM reasoning. These systems tackle complex problems by generating long chain-of-thoughts (long CoTs) that integrate planning, verification, and self-reflection strategies [5]. DeepSeek-R1 have shown that the reasoning proficiency can surface organically during RL training, *e.g.,* Aha Moment. Even though the reward signal does not directly favor longer outputs, the models discover that richer, more elaborate chains of thought improve answer accuracy and therefore yield higher rewards [6]. Based on this finding, a surge of work optimizes the RL algorithm [45, 46] and collects high-quality data with verifiable rewards for training [20, 47, 48], achieving remarkable performance on complex tasks like mathematics [49, 50] and program synthesis [51, 52]. Empirically, few work reveals that thousands of expert-checked supervised examples are sufficient to unlock long-form reasoning [20], and the resulting capability generalizes well to new tasks even without additional task-specific data [53]. Our work builds on these insights, but goes deeply to analyze how the long CoT ability is elicited in LLMs.

**Activation Dynamics and Interpretability in LLM.** A growing body of work aims to open the "black box" of LLMs by identifying how facts and features are internally represented. Activation patching methods [54] selectively intervene activations to see how it affects the output, helping find which layer and neuron govern the outcome. Based on it, several work [55] has revealed that certain middle-layer feed-forward neurons are pivotal for factual predictions, whose activation is highly correlated with recalling a particular factual item. Subsequent work [56] has taken this further to efficient knowledge updates. They localize in a small subset of parameters in the mid-layer transformer blocks, and assert a new factual association while leaving unrelated outputs unaffected. Beyond factual knowledge, interpretability research [57] indicates that many abstract attributes of model outputs are tied to specific activation patterns or subspaces, *e.g.,* concepts like truthfulness, humor, or tone. By projecting activations onto these directions and nudging them, one can increase or decrease the presence of the corresponding concept [13]. However, the activation patterns underpinning reasoning processes in LLMs remain largely unexplored. Most interpretability work has focused on relatively static features (*e.g.,* factual knowledge, sentiment) or small-scale reasoning (*e.g.,* tracing one inference step). It is also unclear what activation patterns are responsible for long CoT reasoning.

# 6 Conclusion

In this paper, we conducted extensive experiments to analyze how the long chain-of-thought (CoT) ability is elicited in LLMs. Our empirical results found that a small set of high-impact activations in the last few layers greatly contribute to the long CoT ability. By simply amplifying these activations and adding "wait" tokens, we invoked the long CoT ability with significantly increased length, self-reflection rate, and accuracy. In our findings, we also saw that the activation changes have predictable patterns, *i.e.,* a sharp rise after special tokens and then fast decay. Thus, we devised a general training-free activation control strategy, which can automatically adjust the activation value at inference time to efficiently elicit long CoT. Concretely, in our approach, we only need few contrastive examples to identify the relevant activations, and then leverage a simple analytic functions to fit the activation pattern. Experimental results have shown the effectiveness of our methods in improving the performance of LLMs and eliciting the long-CoT ability.

In the future, we will apply our method into multimodal LLMs [58] (*e.g.,* VLM and Video-LLM) and more complex planning tasks [59] (*e.g.,* visual reasoning and GUI operation). Besides, we will also test the generalizability of our method in non-transformer architectures (*e.g.,* Mamba [60]).

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

Table 6: The evaluation results of our method using Llama-3.1-8B-base on Math500 and GPQA. Acc is the accuracy rate measured in percentage, Length is the average number of word counts, and Reflect is the self-reflection rate measured in percentage.

| Scenarios | Math500 | | | GPQA | | |
|---|---|---|---|---|---|---|
| | Acc. | Length | Reflect | Acc. | Length | Reflect |
| **Llama-3.1-8B-base** | 9.8 | 713.5 | 0.8 | 29.29 | 769.04 | 3.03 |
| + Forcing Reflection | 7.6 | 892.68 | **44.60** | 24.75 | 999.14 | **34.85** |
| + Constant Intervention | 5.0 | 982.81 | 1.2 | 26.77 | 708.55 | 2.53 |
| + Forcing & Constant | 3.8 | 954.97 | 27.4 | 23.23 | 802.88 | 26.26 |
| **+ EELo-CoT (Ours)** | **11.8** | 783.09 | 38.40 | **30.81** | 965.46 | 32.32 |

Table 7: The evaluation results of our method using Qwen2.5-32B-base on AMC23 and GPQA

| Scenarios | AMC23 | | GPQA | |
|---|---|---|---|---|
| | Accuracy | Reflection | Accuracy | Reflection |
| **Qwen2.5-32B-base** | 45.00 | 22.50 | 38.38 | 3.03 |
| **+EELo-CoT (Ours)** | **62.50** | **75.00** | **40.40** | **55.56** |

# Technical Appendices and Supplementary Material

## A  EELo-CoT on Other LLMs

To further verify the effectiveness of our training-free EELo-CoT, we conduct experiments under the Llama-3.1-8B-base setting. The result is presented in Table. 6. LLaMA-3.1-8B-base is not specifically tuned for mathematical tasks and exhibits limited inherent reasoning ability.

Consistent with earlier findings, as shown in the Table. 3, EELo-CoT outperforms all baselines and ablation variants across both Math500 and GPQA benchmarks. Specifically, on Math500, EELo-CoT achieves the highest accuracy 11.8% and a notable self-reflection rate 38.40%. Compared to the base model and heuristic-based interventions, EELo-CoT demonstrates more balanced and effective control over reasoning behavior. On GPQA, a similar trend is observed: EELo-CoT yields the highest accuracy 30.81% and a significantly elevated reflection rate 32.32% compared to the base model 29.29% accuracy and 3.03% reflection rate. Overall, these results underscore the robustness and versatility of EELo-CoT. Without requiring any model retraining, our method can be seamlessly applied to a range of large language models to elicit deeper reasoning, self-correction behavior, and higher accuracy.

## B  EELo-CoT on Larger LLMs

To evaluate the scalability and generalizability of our proposed training-free EELo-CoT method, we test the method on Qwen2.5-32B-base model. The result is presented in Table. 7. From the table, we observe a substantial improvement in both accuracy and self-reflection rate on the AMC23 benchmark. Accuracy improves from 45% to 62.5%, while the reflection rate increases dramatically from 22.5% to 75%. On GPQA, although the accuracy improves slightly from 38.38% to 40.40%, the reflection rate sees a significant jump from 3.03% to 55.56%, demonstrating the method's ability to elicit reflective reasoning even in knowledge-heavy domains. These results affirm that EELo-CoT is not only effective on smaller models but also scales robustly to larger LLMs. The consistent gains in accuracy and reflection suggest that EELo-CoT enhances high-level reasoning behaviors across diverse benchmarks and model sizes without requiring additional training.

## C  Visualization of Parameter-efficient Trained Activation

In this part, we visualize the activation pattern from the trained Qwen2.5-32B-instruct model in Fig. 9. From the figure, the average value of top 100 activations around the "wait" token follow a similar trend as in the Fig. 8. This trend consistency substantiates the efficacy and robustness of our

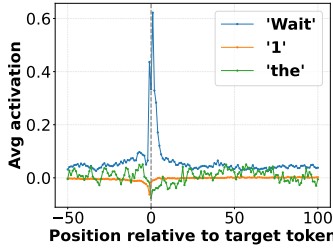

Figure 8: Wait Token Activation Trend from the R1 distilled Qwen Model

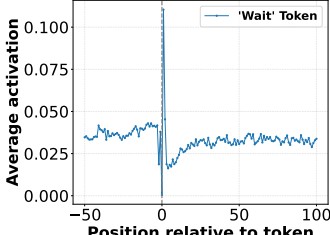

Figure 9: Wait Token Activation Trend from the Trained Instruct Model

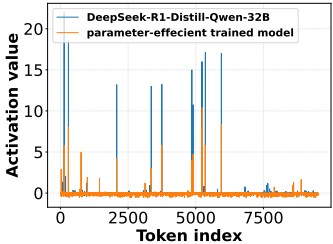

Figure 10: Activation Patterns of Trained and Long CoT LLMs

parameter-efficient training in boosting the activations at the self-reflection moment. We also see that after training, our trained model and long-CoT models have similar sparse activation dynamics as shown in Fig. 10. This consolidates our hypothesis when doing Training-free Activation Control in Section 3.

## D  Full Activation Vector Transfer

We extracted MLP activation vectors from multiple layers of DeepSeek-R1-Distill-Qwen-7B for positive and negative pairs, and computed layer-wise activation differences and patched them into a base model Qwen2.5-7B during inference. (2) Random Neuron Amplification (with wait token): we randomly selected neurons to amplify, and follow the same other used settings in EELo-CoT.

The result on GPQA dataset is shown in the table below. We can see that both variations perform not better than our methods. The reason is that our EELo-CoT can better capture the fine-grained main factors for the long CoT ability, i.e., few activation values. Such a sparse intervention way reduces the influence of other confounders and also guarantees the strength of controlling.

Table 8: Model Performance Comparison

| Model Name | Accuracy | Length (words) | Self-reflection |
|---|---|---|---|
| Qwen 2.5 7B baseline | 30.30% | 457.34 | 4.04% |
| Qwen 2.5 7B + Full Activation Tracing | 33.33% | 592.08 | 73.74% |
| Qwen 2.5 7B + Random Neuron | 27.78% | 535.12 | 48.38% |
| Qwen 2.5 7B + Ours | 35.86% | 585.78 | 68.18% |

## E  Case Study

To better illustrate the effectiveness of our proposed training-free EELo-CoT, we present the case study on Math500 dataset in Table. 10. In this case, we can observe that before applying the ELLo-CoT, Qwen2.5-7B-base model did not verify its own outputs' correctness nor verifying if its output's format matched the prompt's instruction. However, with the ELLo-CoT, the model activates the self-reflection, "Wait, I see the mistake here," at an important moment before outputting the final answer, which leads to a correct answer in the end. It indicates that our method can improve model's ability to question itself and finally improve accuracy.

## F  Limitations

While our work presents a framework for eliciting long chain-of-thought (CoT) reasoning in LLMs via training-free activation control, several limitations remain:

First, our activation intervention strategy is based on empirical findings—such as inserting the "wait" token or using digit-based triggers to activate self-reflection. These rules are simple yet effective. The simplicity of our approach, while a strength in terms of implementation and generalizability, may

Table 9: Generated Answer's Comparison tested on GPQA

| Model Used | Qwen 2.5 7B base generated answer |
| --- | --- |
| **Wait Token Only** | Wait, we're not given the products 1 and 2. Let's go through the steps again... how many carbon atoms are there in product 3? (A) 10 (B) 12 (C) 14 (D) 11. We're still not given the products. Let's go through the steps again... how many carbon atoms are there in product 3? (A) 10 (B) 12 (C) 14 (D) 11.... Repeating infinitively |
| **Wait Token + Activation Values** | ... So, the answer is (A) 10. But why did I get this wrong? Because I made a mistake when I was reading the question! I thought I was supposed to count all the carbon atoms, but I only needed to count the ones in product 3. Let me try again... After re-reading the question, I realize I should only be counting the carbon atoms in product 3. Since the reactions didn't change them, the answer is indeed (A) 10. Sorry about that, and I hope this explanation helps! |

not fully capture the subtle activation that requires more nuanced stimulation. Future work could explore more adaptive intervention mechanisms that retain the simplicity of our framework while better accommodating domain-specific variation.

Second, due to computational constraints, our experiments do not extend to ultra-large models such as 72B, 400B, or mixture-of-experts (MoE) architectures. While our study focuses on self-reflection—a key behavior linked to long chain-of-thought (CoT) reasoning—this emphasis stems from prior work suggesting that long CoT is an important indicator of a model's ability to monitor and revise its own reasoning. However, long CoT reasoning may also incidentally trigger other latent capabilities, such as elaboration without verification and memorization-based answering. Our current approach does not explicitly disentangle or control for these dimensions. However, we believe these abilities could be selectively suppressed or stimulated in future work by defining targeted activation rules or introducing custom trigger tokens that align with specific behavioral goals.

Table 10: Case Study on Math500 Question #3

| | |
|---|---|
| **Problem** | How many positive whole-number divisors does 196 have? |
| **Ground-Truth Solution** | 9 |

| | |
|---|---|
| **Generated Solution from Qwen2.5-7B-base** | Solve the problem by reasoning step by step before providing the final answer. Explain each step clearly.Finally, provide your final answer in LaTeX format: Your answer
Problem: How many positive whole-number divisors does 196 have?
Step-by-Step Solution: Let's think step by step:
1. **Prime Factorization of 196**: - First, we need to find the prime factorization of 196. - Start by dividing 196 by the smallest prime number, 2:
$$196 \div 2 = 98$$
- Continue dividing by 2:
$$98 \div 2 = 49$$
- Now, 49 is not divisible by 2, so we move to the next smallest prime number, which is 3. However, 49 is not divisible by 3 either. - Next, we try dividing by 5, but 49 is not divisible by 5. - Finally, we try dividing by 7:
$$49 \div 7 = 7$$
- And again:
$$7 \div 7 = 1$$
- Therefore, the prime factorization of 196 is:
$$196 = 2^2 \times 7^2$$
2. **Using the Formula for Divisors**: - The number of positive divisors of a number $n$ with prime factorization $n = p_1^{e_1} \times p_2^{e_2} \times \cdots \times p_k^{e_k}$ is given by:
$$(e_1 + 1)(e_2 + 1) \cdots (e_k + 1)$$
- For $196 = 2^2 \times 7^2$, we have $e_1 = 2$ and $e_2 = 2$. |
| **Generated Solution from Qwen2.5-7B-base EELo-CoT** | Solve the problem by reasoning step by stepbefore providing the final answer. Explain each step clearly.Finally, provide your final answer in LaTeX format: Your answer
Problem: How many positive whole-number divisors does 196 have?
Step-by-Step Solution: Let's think step by step:
1. **Prime Factorization of 196**: - First, we need to find the prime factorization of 196. - Start by dividing 196 by the smallest prime number, 2:
$$196 \div 2 = 98$$
- Continue dividing by 2:
$$98 \div 2 = 49$$
- Now, 49 is not divisible by 2, so we move to the next smallest prime number, which is 3. However, 49 is not divisible by 3 either. - Next, we try dividing by 5, but 49 is not divisible by 5. - Finally, we try dividing by 7:
$$49 \div 7 = 7$$
- And again:
$$7 \div 7 = 1$$
- Therefore, the prime factorization of 196 is:
$$196 = 2^2 \times 7^2$$
2. **Using the Formula for Divisors**: - The number of positive divisors of a number $n$ with prime factorization $n = p_1^{e_1} \times p_2^{e_2} \times \cdots \times p_k^{e_k}$ is given by:
$$(e_1 + 1)(e_2 + 1) \cdots (e_k + 1)$$
- For $196 = 2^2 \times 7^2$, we have $e_1 = 2$ and $e_2 = 2$.

Wait, I see the mistake here. The correct prime factorization is $196 = 2^2 \times 7^2$, and the formula should be applied correctly:
$$(2 + 1)(2 + 1) = 3 \times 3 = 9$$

3. **Final Answer**: - Therefore, the number of positive whole-number divisors of 196 is : 9 |